## RESEARCH ARTICLE

# Respiratory syncytial virus burden among Ugandan adults aged ≥65 years: A 15-year sentinel surveillance study of prevalence, coinfections, and comorbidities (2010–2025)

Haruna Muwonge[1,2]*, Joyce Namulondo[3], Levicatus Mugenyi[4], Joweria Nakaseegu[3], Bridget Nakamoga[3], Esther Amwine[3], Roselyne Akugizibwe[2], Abdul Ssekandi[1], Mustafa Ssaka[1,5,6], Hassan Kasujja[5,6], Godfrey S. Bbosa[1], David Odongo[3], Julius Lutwaama[3], John Kayiwa[3], Bruce Kirenga[1,2], Barnabas Bakamutumaho[2,3]

1 College of Health Sciences, Makerere University, Kampala, Uganda, 2 Interdisciplinary Consortium for Epidemics Research (ICER), Kampala, Uganda, 3 Department of Arbo virology, Emerging and Re-emerging Infectious Diseases, Uganda Virus Research Institute, Entebbe, Uganda, 4 MRC/UVRI & LSHTM Uganda Research Unit, Entebbe, Uganda, 5 Clarke International University, Kampala, Uganda, 6 Habib Medical School, Islamic University in Uganda, Kampala, Uganda

* harunamuwonge@gmail.com

## Abstract

### Background

Respiratory syncytial virus is an important cause of acute respiratory illness in older adults, yet data from sub-Saharan Africa remain scarce. Better understanding of its prevalence, coinfections, seasonal patterns, and risk factors in older populations is essential for strengthening surveillance systems and informing prevention strategies in low-resource settings.

### Methods

We conducted a retrospective cross-sectional study using Uganda's national influenza-like illness and severe acute respiratory infection sentinel surveillance data from December 2010 to January 2025. Adults aged ≥65 years with real-time-PCR results for respiratory syncytial virus, influenza A and B, and SARS-CoV-2 were included. Descriptive analyses summarized prevalence, clinical characteristics, and temporal trends. Poisson regression with robust variance estimated adjusted prevalence ratios for factors associated with infection and hospitalization.

### Results

Among 545 illness episodes (mean age 73.2 years; 54.1% female), the period prevalence of respiratory syncytial virus across 2010–2025 was 4.8% (95% CI 3.3–6.9), comparable to influenza A (4.2%) and lower than SARS-CoV-2 (6.4%). Most RSV cases were mono-infections (92.3%), with rare respiratory syncytial virus–influenza

**Data availability statement:** All relevant data are within the manuscript and its Supporting information files.

**Funding:** Pfizer Inc. through an Investigator Sponsored Research Grant supported this study (Grant Number: 89490323). HM is also currently supported by the Fogarty International Center of the National Institutes of Health under Award Number D43TW010526. The content of this manuscript is solely the responsibility of the authors and does not necessarily represent the official views of the Pfizer Inc. or the National Institutes of Health. The funders had no role in study design, data collection and analysis, decision to publish, or preparation of the manuscript.

**Competing interests:** The authors have declared that no competing interests exist.

coinfections (0.4%) and no respiratory syncytial virus–SARS-CoV-2 coinfections. Asthma and pneumonia were independent predictors of infection. Hospitalization was strongly associated with asthma, pneumonia, and heart disease. Activity showed seasonal peaks in March and June, with a marked decline during 2020–2021 and resurgence thereafter.

## Conclusions

RSV is a consistent contributor to medically attended respiratory illness among older Ugandan adults, with prevalence similar to influenza A. Although strong associations with asthma and pneumonia were observed, asthma-related findings are based on few cases and should be considered hypothesis-generating. Seasonal clustering and post-pandemic resurgence support integrating respiratory syncytial virus into routine respiratory surveillance and inform the targeted introduction of preventive interventions, including vaccination, in low- and middle-income settings.

## Introduction

Respiratory Syncytial Virus (RSV) is an increasingly recognized cause of acute lower respiratory tract infection and severe respiratory disease among older adults, particularly those aged ≥65 years. Recent global analyses demonstrate that RSV contributes substantially to medically attended illness, hospitalization, and mortality in this age group, with the highest burden observed among adults aged ≥75 years and those with underlying cardiopulmonary disease or immunocompromising conditions [1–5]. In high-income settings, RSV-associated hospitalization rates in older adults are now comparable to—and in some contexts exceed—those attributed to seasonal influenza, with estimates of approximately 20–30 hospitalizations per 10,000 persons annually among adults aged ≥65 years [3,5]. Clinical outcomes are frequently severe: nearly half to two-thirds of laboratory-confirmed RSV infections in older adults require inpatient care, and RSV has been associated with prolonged hospital stays, functional decline, and excess all-cause mortality [2,4–6]. These findings have shifted the understanding of RSV from a predominantly pediatric pathogen to a major cause of serious respiratory illness in aging populations.

The burden of RSV in older adults is driven by immunosenescence, characterized by age-related declines in innate and adaptive antiviral immune responses, and is further amplified by the high prevalence of chronic comorbidities such as asthma, chronic obstructive pulmonary disease, cardiovascular disease, diabetes, and chronic kidney disease [2,7–11]. Population-based studies from high-income countries estimate RSV incidence at approximately 600 cases per 100,000 person-years among adults aged ≥60 years, with hospitalization rates exceeding 200 per 100,000 person-years among those with underlying conditions [12,13]. However, evidence from low- and middle-income countries (LMICs), particularly in sub-Saharan Africa, remains sparse and fragmented. Available regional estimates suggest relatively low apparent RSV prevalence among adults, but these findings are largely derived from

non–age-stratified surveillance and are likely to underestimate the true burden in older adults due to limited diagnostic testing, reliance on fever-based case definitions, and barriers to healthcare access [14,15]. As populations in LMICs undergo rapid demographic aging, the absence of robust, age-specific RSV data represents a critical gap for public health planning and clinical preparedness.

In Uganda, the Ministry of Health has implemented a nationwide health facility–based sentinel surveillance system for influenza-like illness (ILI) and severe acute respiratory infection (SARI) since 2010, incorporating polymerase chain reaction (PCR) testing for multiple respiratory viruses and, more recently, SARS-CoV-2 [16]. Analyses from this platform have predominantly focused on pediatric populations or influenza trends, leaving RSV epidemiology among older adults largely understudied. This gap is particularly concerning given that Uganda's population aged ≥60 years is projected to double by 2050, with important implications for respiratory disease burden and geriatric healthcare demand. The availability of a 15-year surveillance window spanning the pre– and post–COVID-19 eras offers a unique opportunity to characterize RSV epidemiology in older adults, including long-term trends and seasonality, in a tropical African setting. This is especially timely in light of the recent approval of three RSV vaccines for older adults—Arexvy (GSK), Abrysvo (Pfizer), and mResvia (Moderna)—all of which received regulatory approval beginning in 2023 following evidence of efficacy against RSV-associated lower respiratory tract disease and hospitalization in adults aged ≥60 years [17,18].

Against this backdrop, we conducted a retrospective analysis of Uganda's Ministry of Health sentinel ILI/SARI data to estimate RSV prevalence among adults aged ≥65 years, describe coinfection patterns with influenza A/B and SARS-CoV-2, investigate associations between RSV positivity with presenting symptoms including selected comorbidities and hospitalization, and examine temporal trends spanning 2010–2025.

## Methods

### Study design

We conducted a retrospective cross-sectional study using routinely collected health facility-based data from Uganda's national influenza-like illness (ILI) and severe acute respiratory infection (SARI) sentinel surveillance system. The analysis covered the period from December 01, 2010 through January 31, 2025. The data were accessed for research purposes on July 01, 2025.

### Surveillance network and study setting

The national sentinel surveillance network comprises 16 public and private healthcare facilities across 11 districts in Uganda, representing diverse ecological and demographic contexts including urban, peri-urban, and rural areas. Facilities included regional referral hospitals, general hospitals, and health centers at levels III and IV. Sites were selected to ensure geographical representativeness. Surveillance involved routine enrolment of outpatients meeting the WHO definition for ILI (an acute respiratory infection with measured fever of ≥38 °C and cough, with onset within the last 10 days) and hospitalized patients meeting SARI criteria (an acute respiratory infection with a history of fever or measured fever of ≥38 °C and cough, with onset within the last 10 days, requiring hospitalization). Enrollment procedures, case definitions, specimen collection protocols, and reporting tools were standardized across all sites and years through national surveillance guidelines, routine staff training, and centralized supervision. Importantly, enrollment was conducted continuously throughout the calendar year with no season-specific quotas or restrictions; thus, differences observed between wet and dry seasons reflect underlying presentation and healthcare utilization patterns rather than differential surveillance practices. Clinical and laboratory data from participating facilities were centrally collated at the Uganda Virus Research Institute (UVRI) National Influenza Centre (NIC), a WHO-accredited influenza laboratory.

## Study population and eligibility criteria

The study population consisted of adults aged 65 years or older, a threshold commonly used to define older adults in RSV epidemiology and vaccine policy, as RSV-related morbidity, hospitalization, and mortality increase sharply beyond this age. Eligible participants presented to sentinel health facilities with symptoms meeting WHO-defined criteria for ILI or SARI. Nasopharyngeal or oropharyngeal specimens were collected and tested for respiratory pathogens, including RSV, influenza sub-type A and B, and SARS-CoV-2. We excluded individuals with missing demographic or laboratory data, repeat visits within a 14-day window for the same illness, individuals vaccinated against RSV or influenza during the study period, those with severe immunosuppressive conditions, and patients presenting with respiratory symptoms due to non-infectious etiologies.

## Data collection and study variables

Clinical and demographic data collected included age, sex, clinical symptoms (e.g., cough, fever, sore throat, shortness of breath, fatigue, and diarrhoea), underlying comorbidities (hypertension, diabetes, heart disease, HIV infection, asthma, active or prior tuberculosis, and cancer), and hospitalization status. Pneumonia was captured as a clinician-diagnosed condition at the point of care based on routine clinical assessment recorded on sentinel case investigation forms; systematic radiographic confirmation or standardized ICD coding was not consistently available across sites. Laboratory data were matched to clinical records using unique patient identifiers. The primary study outcome was laboratory-confirmed RSV infection by real-time reverse transcription polymerase chain reaction (RT-PCR). Secondary outcomes included coinfection with Influenza sub-types A and B and SARS-CoV-2, seasonal trends, and associations between RSV positivity and symptoms and severity or comorbid conditions. All analyses were conducted at the level of individual medically attended ILI/SARI illness episodes rather than unique patients. Because surveillance enrollment is conditional on meeting ILI/SARI criteria, prevalence estimates reflect test positivity among distinct illness episodes in age-eligible adults rather than community-level incidence.

## Laboratory procedures

Nasopharyngeal and/or oropharyngeal swabs were collected using sterile Dacron swabs, placed in viral transport medium, and stored at 4°C at sentinel facilities. Samples were shipped bi-weekly to UVRI-NIC under standardized cold-chain conditions. At UVRI-NIC, total nucleic acid was extracted using the Andis Viral RNA Extraction Kit. RSV detection was performed using the Healgen RSV real-time RT-PCR assay, and influenza A/B and SARS-CoV-2 were detected using the CDC Flu SC2 multiplex RT-PCR assay run on the ABI 7500 PCR platform. Samples with cycle threshold (Ct) values below 40 for RSV and SARS-CoV-2, and below 36 for influenza viruses, were considered positive. Quality control measures included extraction controls, positive and negative amplification controls, and human RNase P internal controls in all test runs. UVRI-NIC participates annually in WHO external quality assessment programmes and maintains ISO-accredited laboratory practices.

## Data management

All surveillance data were anonymized, and securely stored in a password-protected database accessible only by authorized investigators. Data quality assurance included routine checks for missing, duplicate, or inconsistent entries and random verification against original case investigation forms. Data management adhered to the requirements of Uganda's Data Protection and Privacy Act [19].

## Statistical analysis

All analyses were conducted using Stata version 18 (StataCorp, College Station, TX, USA). Descriptive statistics summarized baseline characteristics of participants, with means and standard deviations (SD) reported for continuous variables

and frequencies with percentages for categorical variables. All percentages and prevalence estimates are reported per illness episode. The prevalence of RSV was calculated as proportions with 95% confidence intervals (CI). RSV prevalence was further stratified by demographic factors, clinical presentation, comorbidities, season, and surveillance period (pre- vs. post-COVID). Seasonal analyses compared wet (March–May; September–November) and dry (June–August; December–February) periods.

Comparisons of RSV prevalence with influenza A/B and SARS-CoV-2 were descriptive and intended to provide epidemiological context; no formal pairwise statistical tests were performed to compare prevalence across viruses. Between-group comparisons were performed using Pearson chi-square test or Fisher's exact test where appropriate.

To identify factors associated with RSV positivity, prevalence ratios (PRs) and 95% CIs were estimated using random effects Poisson regression models with robust variance. The random effects models were considered to account for within site clustering. Variables were considered for inclusion in multivariable models based on biological plausibility and evidence of association using Wald test statistic. Biologically plausible variables were fixed a priori and were retained in the multivariable models regardless of their effect at bivariable level analysis. Demographic variables (e.g., age and sex) were assessed for their contribution in the multivariable random effects Poisson models using the Wald test statistic and were dropped if they did not improve model fit (p > 0.1).

Seasonal dynamics were assessed by plotting monthly positivity rates for RSV and comparator viruses and by comparing aggregate prevalence between wet and dry rainfall periods. Temporal trends were described across the pre-pandemic, pandemic, and post-pandemic phases of surveillance. For severity analyses, hospitalization was used as a proxy outcome for severe RSV illness. Associations between comorbidities and hospitalization among RSV-positive participants were examined using a random effects Poisson regression model with robust variance to estimate adjusted prevalence ratios (aPRs). Multiple imputation by chained equations was used to impute missing data. Results for aPRs are presented for both the complete-case and imputed data analyses. A p-value <0.05 was considered statistically significant.

### Ethical considerations

Ethical approval for this retrospective analysis was obtained from the Makerere University School of Biomedical Sciences Research and Ethics Committee (SBS-REC, approval # SBS-2024–685) and the Uganda National Council for Science and Technology (UNCST, approval # HS5423ES). Given the retrospective nature of anonymized surveillance data, the SBS-REC provided a waiver of individual informed consent. The sentinel surveillance database was accessed for research purposes on July 01, 2025. The database contained only de-identified information, and authors had no access to information that could identify individual participants during or after data collection. All data handling and analyses were conducted in compliance with Uganda's Data Protection and Privacy Act [19].

## Results

### Study cohort profile

Between 1 December 2010 and 31 January 2025 the sentinel network recorded 545 illness episodes in adults aged ≥65 years that satisfied the WHO ILI/SARI case definitions and had a full respiratory-virus RT-PCR panel (RSV, Influenza subtype A and B and SARS-CoV-2). All 545 individual records met the analytical inclusion criteria, so the final study cohort represents 100% of age-eligible surveillance encounters during the study window (Fig 1).

Completeness of the core exposure and outcome variables was high: viral PCR results, age, sex and reporting month were available for every participant. Most symptom fields showed <3% missing data (e.g., cough 0.6%, sore throat 2.4%, shortness of breath 0.4%). In contrast, several comorbidity variables were absent in 14–15% of records (heart disease, hypertension, diabetes, asthma, active and prior tuberculosis, cancer). Hospitalization status was the single variable with substantial missingness, absent in 22.2% of cases, while history of fever was missing in 11.4%. Thus, missing data were largely confined to secondary clinical covariates and are unlikely to bias virological prevalence estimates.

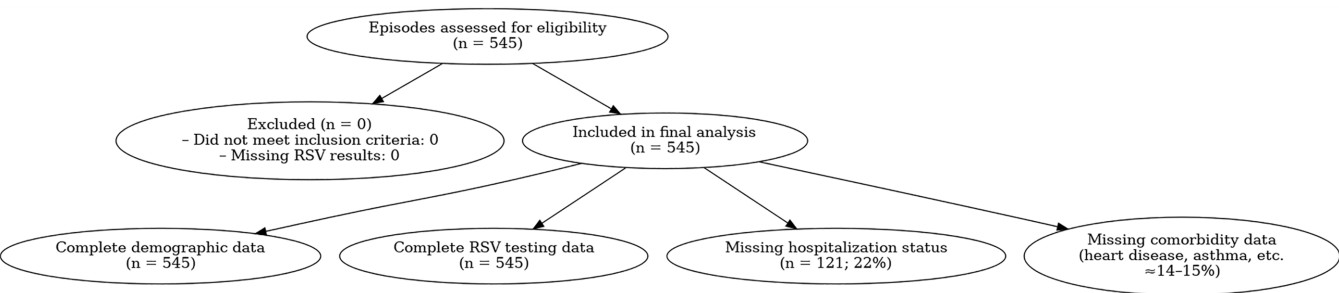

**Fig 1. CONSORT-style flow diagram of surveillance episode inclusion and data completeness.**

## Baseline characteristics of participants

This cohort comprised 545 illness episodes in adults aged ≥65 years. The mean age was 73.2 ± 9.1 years, and women constituted a slight majority (54.1%). Current smoking was uncommon (2.0%). Cough (73.9%), fever or history of fever (71.4%), and shortness of breath (47.5%) were the most frequently reported symptoms, whereas sore throat was present in 27.7% of cases. Documented comorbidities were infrequent but clinically relevant: hypertension (9.5%), diabetes (5.9%), heart disease (3.9%), and HIV infection (3.1%); asthma, active tuberculosis, prior tuberculosis, and cancer each affected < 2% of participants. Nearly two-thirds of encounters (65.7%) resulted in hospital admission, although length-of-stay data were not systematically recorded. Annual enrolment counts increased sharply after 2020, reflecting the scale-up of sentinel testing during the COVID-19 period and subsequent expansion of the surveillance network. Table 1 describes the baseline characteristics of study participants.

## Prevalence of RSV and other respiratory viruses

Over the 15-year surveillance period, 26 of 545 illness episodes tested positive for RSV, yielding an overall period prevalence of 4.8% (95% CI: 3.3–6.9). Fig 2 shows the geographical distribution of confirmed RSV cases across several districts in Uganda. For comparison, the observed period prevalence was 6.4% for SARS-CoV-2, 4.2% for influenza sub-type A, and 1.3% for Influenza sub-type B (Table 1). Table 2 provides a detailed breakdown of RSV prevalence stratified by demographic group, clinical presentation, and comorbidities.

The map was created by the authors using QGIS and primary study data, with open-source base layers used for geographic reference. No proprietary or copyrighted map or satellite imagery was used. The figure is published under the CC BY 4.0 license.

RSV was slightly higher among women (5.8%) than men (3.6%), but the sex-based difference was not statistically significant (p = 0.238). Stratification by surveillance period showed RSV period prevalence declined from 7.1% pre-COVID (2010–2019) to 4.2% post-COVID (2020–2025), though the difference did not reach significance (p = 0.196). RSV period prevalence was also slightly higher in the dry season (5.3%) compared to the wet season (4.3%), again with no significant difference (p = 0.585).

When examined by age, RSV period prevalence was 4.3% among adults aged 65–69, 4.2% among those 70–79, and 6.1% among those aged 80 and above (p = 0.667), showing no clear age-related significant difference. Smoking was not significantly associated with RSV (9.1% vs 4.6%, p = 0.487), though this analysis was limited by small numbers of current smokers.

Among presenting symptoms, cough and sore throat were the only significant correlates of RSV infection. Participants with cough had an RSV positivity of 6.2%, compared to 0.7% among those without cough (p = 0.009). Similarly, sore throat was reported in 7.9% of RSV-positive cases versus 3.1% of RSV-negative cases (p = 0.016). Other symptoms—including

**Table 1. Baseline characteristics of study participants.**

| Characteristic | n (% of cohort*) |
|---|---|
| **Demographics** | |
| Age, mean±SD (years) | 73.21±9.09 |
| Male sex | 250 (45.9) |
| Female sex | 295 (54.1) |
| **Risk behavior** | |
| Current smoker | 11 (2.0) |
| **Presenting symptoms** | |
| Cough | 403 (73.9) |
| Fever or history of fever | 389 (71.4) |
| Shortness of breath | 259 (47.5) |
| Sore throat | 151 (27.7) |
| Headache | 269 (49.4) |
| Muscle pains | 218 (40.0) |
| Nausea | 124 (22.8) |
| Vomiting | 92 (16.9) |
| Diarrhoea | 50 (9.2) |
| **Comorbidities†** | |
| Hypertension | 52 (9.5) |
| Diabetes mellitus | 32 (5.9) |
| Heart disease | 21 (3.9) |
| HIV infection | 17 (3.1) |
| Asthma | 4 (0.7) |
| Tuberculosis (active and prior TB) | 7 (1.3) |
| Cancer | 2 (0.4) |
| **Level of care** | |
| Hospitalized | 358 (65.7) |
| Not hospitalized | 66 (12.1) |
| Hospitalization status missing | 121 (22.2) |
| **Annual enrolment** | |
| 2010–2015 | 70 (12.8) |
| 2016–2019 | 49 (9.0) |
| 2020–2021 | 88 (16.1) |
| 2022–2023 | 241 (44.2) |
| 2024–2025 | 97 (17.8) |
| **Viral infections** | |
| RSV-positive | 26 (4.8) |
| SARS-CoV-2 positive | 35 (6.4) |
| Influenza A positive | 23 (4.2) |
| Influenza B positive | 7 (1.3) |

*Percentages use the total cohort (N=545) as denominator.

†Comorbidity variables were missing in 14–15% of records; proportions are calculated on the full cohort for transparency.

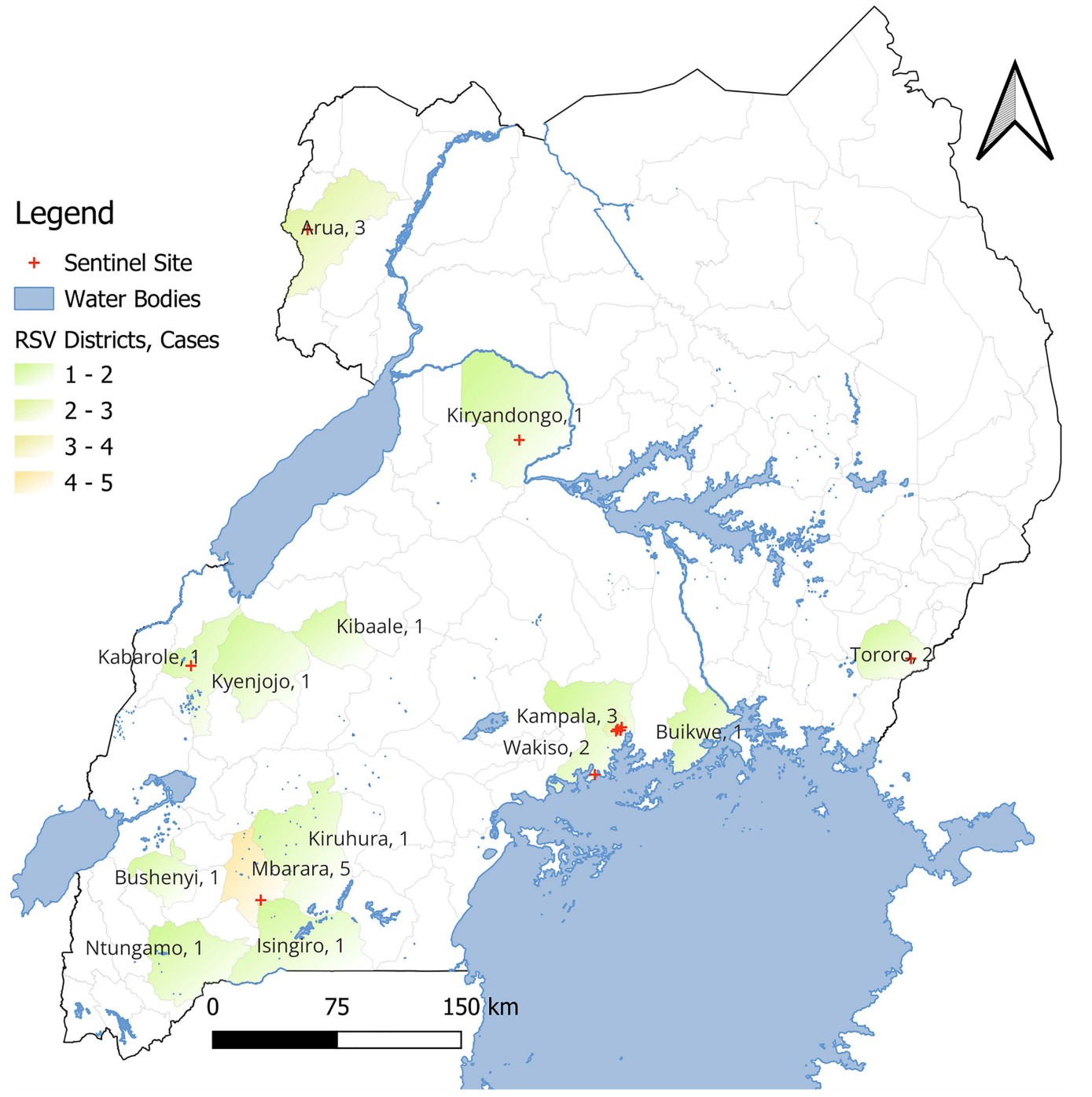

**Fig 2. Geographical distribution of RSV-positive cases among adults aged ≥ 65 years in Uganda, 2010–2025.**

shortness of breath (5.8% vs 3.9%, p = 0.296), fever (5.1% vs 4.3%, p = 0.723), headache, nausea, vomiting, muscle pains, and diarrhoea—showed no statistically significant differences with RSV positivity.

Several comorbid conditions were explored for association with RSV positivity. Asthma emerged as the strongest predictor, with 2 of 4 individuals (50.0%) testing RSV-positive compared to 5.0% among those without asthma (p < 0.001).

**Table 2. RSV prevalence by demographic, clinical, and comorbidity characteristics among adults aged ≥65 years.**

| Characteristic | RSV-positive n/N (%) | 95% CI | p-value* |
|---|---|---|---|
| **Sex** | | | |
| Male | 9/250 (3.6) | 1.9–6.8 | 0.238 |
| Female | 17/295 (5.8) | 3.6–9.1 | |
| **COVID period‡** | | | |
| Pre-COVID | 8/113 (7.1) | 3.6–13.8 | 0.196 |
| Post-COVID | 18/432 (4.2) | 2.6–6.6 | |
| **Season** | | | |
| Dry | 14/265 (5.3) | 3.2–8.8 | 0.585 |
| Wet | 12/280 (4.3) | 2.5–7.5 | |
| **Age group** | | | |
| 65–69 | 9/209 (4.3) | 2.3–8.2 | 0.667 |
| 70–79 | 8/189 (4.2) | 2.1–8.4 | |
| 80+ | 9/147 (6.1) | 3.2–11.6 | |
| **Smoking status** | | | |
| Yes | 1/11 (9.1) | 1.3–64.5 | 0.487 |
| No | 24/521 (4.6) | 3.1–6.8 | |
| **Symptoms** | | | |
| Cough | 25/403 (6.2) | 4.2–9.1 | **0.009** |
| Sore throat | 12/151 (7.9) | 4.6–13.7 | **0.016** |
| Shortness of breath | 15/259 (5.8) | 3.5–9.5 | 0.296 |
| Fever/history of fever | 20/389 (5.1) | 3.4–7.9 | 0.723 |
| Vomiting | 6/92 (6.5) | 3.0–14.2 | 0.390 |
| Headache | 12/269 (4.5) | 2.6–7.8 | 0.716 |
| Nausea | 6/124 (4.8) | 2.2–10.6 | 0.972 |
| Muscle pains | 14/218 (6.4) | 3.9–10.7 | 0.144 |
| Diarrhoea | 5/50 (10.0) | 4.3–23.2 | 0.070 |
| **Comorbidities** | | | |
| Asthma | 2/4 (50.0) | 16.1–1.0 | **<0.001** |
| Heart disease | 3/21 (14.3) | 4.9–41.8 | 0.063 |
| Pneumonia† | 6/62 (9.7) | 4.5–20.8 | 0.054 |
| TB (active + prior) | 1/13 (7.7) | 1.4–71.0 | 0.736 |
| Diabetes mellitus | 2/32 (6.2) | 1.6–24.4 | 0.686 |
| Hypertension | 4/52 (7.7) | 3.0–19.9 | 0.299 |
| HIV | 0/17 (0.0) | --- | 0.348 |
| Cancer | 0/2 (0.0) | -- | 0.736 |

*Pearson χ² or Fisher's exact test as appropriate. CI = confidence interval.*

† *Diagnosis recorded by admitting clinician.*

‡ *Pre-COVID period defined as January 2010–February 2020; post-COVID period defined as March 2020–January 2025, corresponding to the onset of SARS-CoV-2 transmission and COVID-19–related public health measures in Uganda.*

Heart disease (14.3% vs 4.9%, p = 0.063) and pneumonia noted by the admitting clinician (9.7% vs 4.1%, p = 0.054) modest significance. No associations were detected between RSV and diabetes mellitus, hypertension, tuberculosis (active or prior), HIV, or cancer.]

**Temporal trends and seasonal dynamics of RSV and comparator viruses**

**Annual trends and pandemic-associated disruptions.** As shown in Fig 3, RSV circulation was observed in most years from 2011 to 2025, with peaks in 2012, 2016, and 2023. RSV detections dropped sharply in 2020 and 2021, coinciding with the COVID-19 pandemic and national restrictions. During the same period, SARS-CoV-2 emerged as the dominant respiratory virus, peaking at ~17% in 2021. Influenza A activity was markedly reduced but rebounded in 2023, while influenza B showed irregular, low-level circulation, with its highest peak in 2013.

Pre- and post-pandemic stratification (Fig 4) shows RSV period prevalence declined from 7.1% before COVID-19 (2010–2019) to 4.2% afterwards (2020–2025). Influenza A and B also showed reduced activity during the pandemic, with subsequent resurgence from 2022 onwards.

**Monthly patterns and seasonality.** Higher-resolution monthly data, shown in Fig 5, reveals the underlying seasonality of RSV, with consistent peaks in May and June; closely aligned with the transition period between the end of the long rainy season and the onset of the short dry season in Uganda.

In contrast, SARS-CoV-2 exhibited classic wave-like behaviour, with sharp surges in January and May–June 2021. Influenza A demonstrated less predictable seasonality, with variable peaks in March–April, while influenza B showed sporadic circulation without a discernible seasonal signature.

**Seasonal variation in virus positivity by rainfall period.** To evaluate seasonal associations, months were classified into wet seasons (March–May and September–November) and dry seasons (June–August and December–February), following Uganda's bimodal rainfall pattern. As illustrated in Fig 6, RSV period prevalence during dry months was 5.3%, slightly higher than the 4.3% observed in wet months. However, this difference was modest and not statistically significant (p = 0.585). For comparison, SARS-CoV-2 showed a notable seasonal gradient, with dry-season positivity of 8.3% versus 4.6% in the wet season. This was largely driven by a pronounced peak during the January 2021 wave. Conversely, influenza A appeared more common in dry months (4.9%) than in wet months (3.6%), though the difference was small. These patterns suggest that while RSV exhibits some seasonal variation in older Ugandan adults, the association with rainfall is weaker than previously hypothesized. SARS-CoV-2 and influenza also demonstrated heterogeneous seasonal behavior, likely influenced by pandemic dynamics and public health interventions.

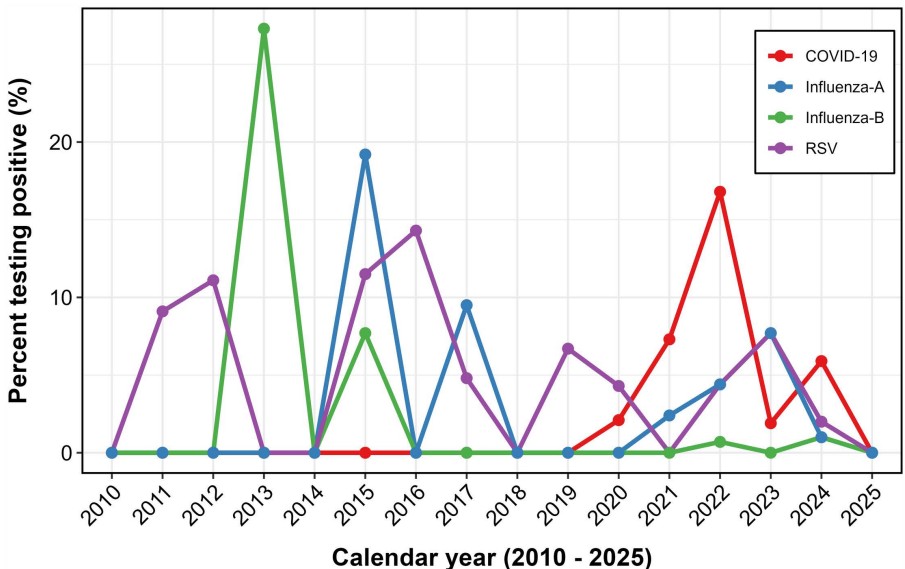

**Fig 3. Annual positivity trends of RSV, influenza A/B, and SARS-CoV-2 among adults' ≥ 65 years in Uganda, 2010–2025.**

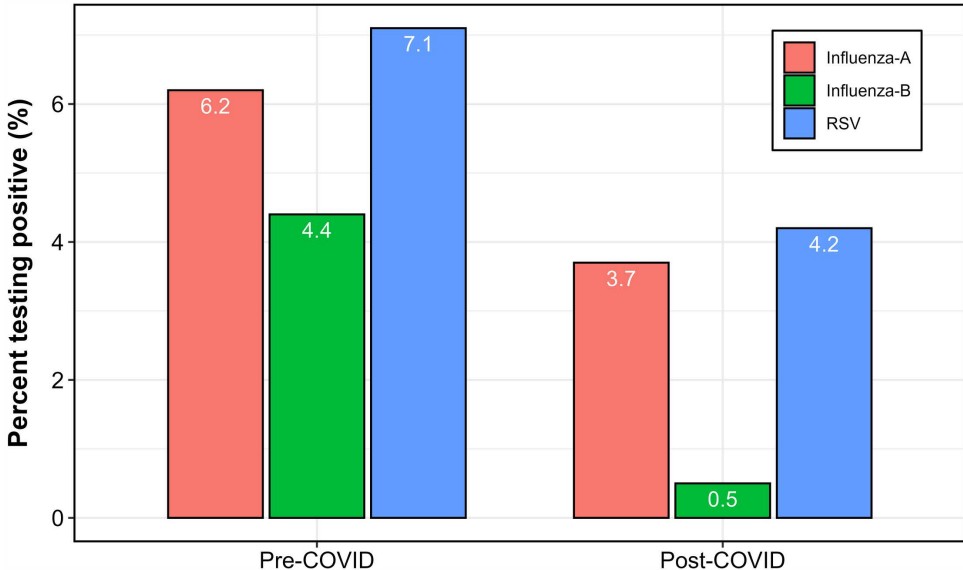

**Fig 4. Pre- and post-COVID period prevalence of RSV, influenza A, and influenza B among adults ≥65 years in Uganda.**

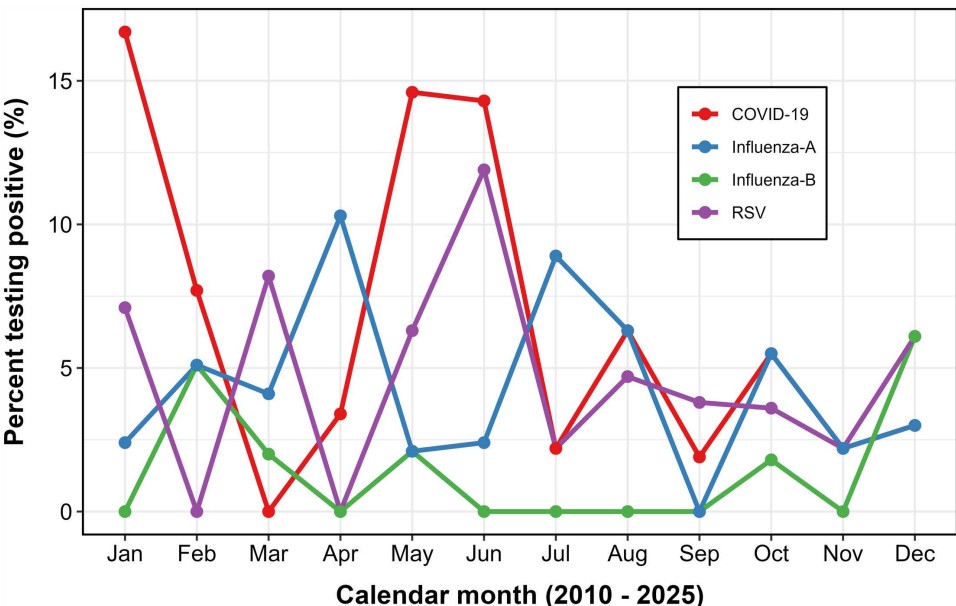

**Fig 5. Monthly positivity trends of respiratory viruses among adults aged ≥65 years, Uganda, 2010–2025.**

### RSV multi-infection patterns

Among the 545 illness episodes, 26 (4.8%) tested positive for RSV. Of these, 24 (92.3%) were RSV mono-infections, while 2 (0.4%) represented coinfections with influenza viruses. No instances of RSV–SARS-CoV-2 coinfection were identified. An additional 63 cases (11.6%) were positive for SARS-CoV-2 and/or influenza A or B in the absence of RSV, whereas 456 participants (83.7%) had no detectable viral pathogen.

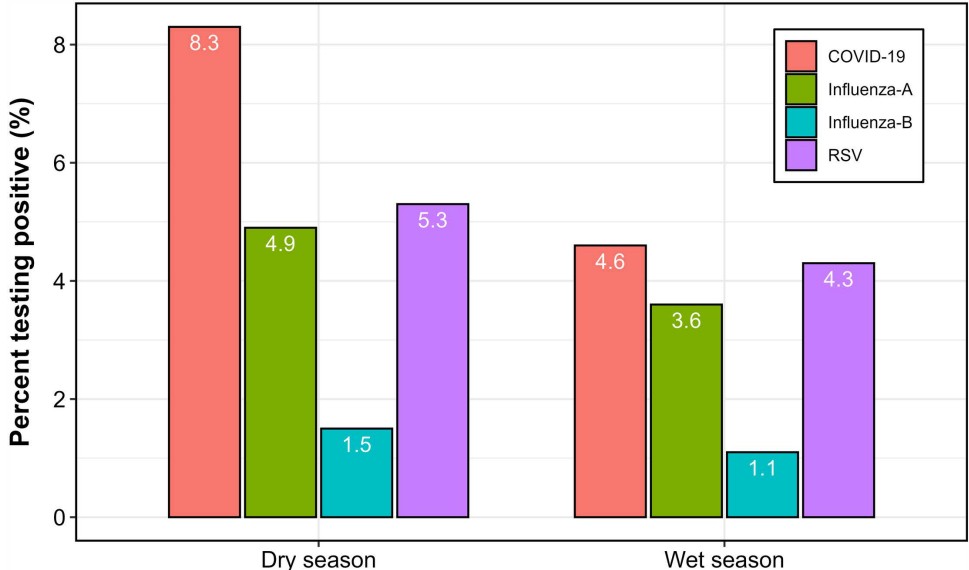

**Fig 6. Seasonal period prevalence of respiratory viruses among adults' ≥ 65 years in Uganda, 2010–2025.**

Stratified analysis revealed several key differences across infection categories (Table 3). RSV positivity (either alone or with influenza) was more common among women (5.7%) than men (3.6%), but the difference was not statistically significant (p = 0.217). Current smokers had numerically higher rates of RSV-only infection (9.1%) and non-RSV viral infection (27.3%), but this was based on small numbers and did not reach significance (p = 0.318).

Cough and sore throat were the only clinical symptoms significantly associated with RSV detection. Among participants with RSV-only infection, 95.8% reported cough and 45.8% reported sore throat; corresponding values among those coinfected with influenza were 100% and 50.0%, respectively. This contrasted with significantly lower frequencies among virus-negative cases (71.5% and 25.0%, respectively), yielding p-values of 0.012 for cough and 0.009 for sore throat.

Comorbid conditions were disproportionately represented among coinfected individuals. Half of the RSV–influenza coinfected cases had documented asthma and heart disease, compared to <1% and 3.9% respectively among virus-negative participants. These associations were statistically significant (p < 0.001 for asthma, p = 0.018 for heart disease). Other conditions such as diabetes, hypertension, tuberculosis, and HIV did not differ significantly across infection categories.

## Factors associated with RSV positivity

A multivariable Poisson regression model with robust variance was used to identify factors independently associated with RSV positivity among older adults. The analysis included 451 participants with complete data on all covariates (Table 4).

In bivariate analysis, the presence of cough was associated with a higher RSV period prevalence (PR: 7.37; [95% CI: 0.66–82.50]; p = 0.105), although this association did not reach statistical significance. Similarly, sore throat was associated with increased RSV prevalence in unadjusted analysis (PR: 2.26; [95% CI: 1.25–4.10]; p = 0.007). However, after adjustment for age and sex, neither cough nor sore throat remained statistically significant in either the complete-case model (aPR: 4.97; [95% CI: 0.43–57.35]; p = 0.199 and aPR: 1.71; [95% CI: 0.85–3.41]; p = 0.130, respectively) nor the imputed model.

Among comorbid conditions, asthma and clinician-diagnosed pneumonia showed the strongest associations with RSV positivity, though with important differences by model specification. Asthma was rare in the cohort (n = 4) but was associated with a markedly higher RSV period prevalence (50.0%). In the imputed adjusted model, asthma remained

**Table 3. RSV mono-infection and coinfection profiles by demographic and clinical characteristics in adults aged ≥65 years.**

| Variable (n) | RSV multi-infection n (%) | | | | P-value |
| --- | --- | --- | --- | --- | --- |
| | Only RSV | RSV + Influenza | Other viruses‡ | No Virus Detected | |
| **Gender (n = 545)** | | | | | 0.217 |
| Male | 8 (3.2) | 1 (0.4) | 23 (9.2) | 218 (87.2) | |
| Female | 16 (5.4) | 1 (0.3) | 40 (13.6) | 238 (80.7) | |
| **Smoking (n = 532)** | | | | | 0.318 |
| Yes | 1 (9.1) | 0 (0) | 3 (27.3) | 7 (63.6) | |
| No | 22 (4.2) | 2 (0.4) | 59 (11.3) | 438 (84.1) | |
| **Cough (n = 545)** | | | | | **0.012** |
| Yes | 23 (5.7) | 2 (0.5) | 52 (12.9) | 326 (80.9) | |
| No | 1 (0.7) | 0 (0) | 10 (7.2) | 131 (92.1) | |
| **Sore throat (n = 545)** | | | | | **0.009** |
| Yes | 11 (7.3) | 1 (0.7) | 25 (16.6) | 114 (75.5) | |
| No | 13 (2.9) | 1 (0.3) | 38 (9.7) | 343 (87.1) | |
| **Heart disease (n = 524)** | | | | | **0.018** |
| Yes | 3 (14.3) | 0 (0) | 6 (28.6) | 12 (57.1) | |
| No | 21 (4.5) | 2 (0.4) | 60 (11.9) | 418 (83.1) | |
| **Asthma (n = 521)** | | | | | **<0.001** |
| Yes | 2 (50) | 0 (0) | 2 (50) | 0 (0) | |
| No | 22 (4.5) | 2 (0.4) | 64 (12.3) | 427 (82.7) | |

‡*Influenza A, influenza B and/or SARS-CoV-2 without RSV.*

independently associated with RSV positivity (aPR: 5.82; [95% CI: 1.60–21.12]; p = 0.007), whereas the complete-case adjusted estimate was imprecise and not statistically significant (aPR: 3.67; [95% CI: 0.19–72.87]; p = 0.393).

Participants diagnosed with pneumonia during admission also had an increased RSV period prevalence (9.7%). This association was statistically significant in the complete-case adjusted analysis (aPR: 3.28; [95% CI: 1.36–7.86]; p = 0.008), but attenuated and no longer statistically significant in the imputed model (aPR: 1.99; [95% CI: 0.98–4.04]; p = 0.056).

Heart disease was associated with higher RSV prevalence (14.3% vs. 4.9% among those without heart disease). After adjustment, heart disease showed a borderline association in the complete-case analysis (aPR: 2.21; [95% CI: 0.97–5.06]; p = 0.061) and reached statistical significance in the imputed model (aPR: 2.18; [95% CI: 1.01–4.71]; p = 0.048).

No significant differences in RSV period prevalence were observed by surveillance period or season. Gastrointestinal symptoms such as diarrhoea were not associated with RSV positivity in either unadjusted or adjusted analyses.

## Predictors of severe RSV illness (hospitalization)

Hospitalization was used as a proxy indicator for severe RSV illness. Among 466 participants with complete data, we examined the association between selected comorbid conditions and hospitalization in RSV-positive cases (Table 5).

Asthma was the strongest independent predictor of severe RSV illness across all analytical approaches. Among RSV-positive individuals with asthma, hospitalization prevalence was 50.0%, compared with approximately 5% among those without asthma. In adjusted analyses, asthma remained strongly associated with hospitalization in both the complete-case model (aPR: 21.69; [95% CI: 7.50–62.71]; p < 0.001) and the imputed model (aPR: 10.83; [95% CI: 2.25–54.27]; p = 0.003).

Clinician-diagnosed pneumonia was also associated with increased hospitalization prevalence. This association was statistically significant in the complete-case adjusted analysis (aPR: 3.80; 95% CI: 1.88–7.56; p = 0.020), but was attenuated and did not reach statistical significance in the imputed model (aPR: 2.46; 95% CI: 0.98–6.29; p = 0.055).

**Table 4. Factors associated with RSV positivity among adults aged ≥65 years.**

| Risk factor | RSV prevalence % (95% CI) | Unadjusted PR (95% CI)* | P value | Complete case Adjusted© PR (95% CI)* | P value | Imputed Adjusted© PR (95% CI) | P value |
|---|---|---|---|---|---|---|---|
| **COVID period** | | | | | | | |
| Pre-COVID | 7.1 (3.6, 13.8) | 1.00 | | 1.00 | | 1.00 | |
| Post-COVID | 4.2 (2.7, 6.6) | 0.77 (0.31, 1.91) | 0.574 | 1.17 (0.39, 3.48) | 0.777 | 0.83 (0.37, 1.88) | 0.664 |
| **Season** | | | | | | | |
| Dry | 5.3 (3.2, 8.8) | 1.00 | | 1.00 | | 1.00 | |
| Wet | 4.3 (2.5, 7.5) | 0.68 (0.24, 1.92) | 0.464 | 0.91 (0.36, 2.29) | 0.840 | 0.93 (0.42, 2.04) | 0.852 |
| **Cough** | | | | | | | |
| No | 0.7 (0.1, 5.1) | 1.00 | | 1.00 | | 1.00 | |
| Yes | 6.2 (4.2, 9.1) | 7.37 (0.66, 82.50) | 0.105 | 4.97 (0.43, 57.35) | 0.199 | 5.76 (0.51, 64.91) | 0.157 |
| **Sore throat** | | | | | | | |
| No | 3.1 (1.8, 5.5) | 1.00 | | 1.00 | | 1.00 | |
| Yes | 7.9 (4.6, 13.7) | 2.26 (1.25, 4.10) | 0.007 | 1.71 (0.85, 3.41) | 0.130 | 1.81 (0.85, 3.85) | 0.123 |
| **Diarrhea** | | | | | | | |
| No | 4.3 (2.8, 6.5) | 1.00 | | 1.00 | | 1.00 | |
| Yes | 10.0(4.3,23.2) | 1.60 (0.35, 7.31) | 0.546 | 1.29 (0.28, 5.87) | 0.740 | 1.73 (0.66, 4.55) | 0.269 |
| **Heart disease** | | | | | | | |
| No | 4.9 (3.3, 7.4) | 1.00 | | 1.00 | | 1.00 | |
| Yes | 14.3(4.9, 41.8) | 2.81 (1.32, 5.98) | 0.007 | 2.21 (0.97, 5.06) | 0.061 | 2.18 (1.01, 4.71) | 0.048 |
| **Asthma** | | | | | | | |
| No | 5.0 (3.3, 7.4) | 1.00 | | 1.00 | | 1.00 | |
| Yes | 50.0 (16.1, 99.9) | 2.89 (0.34, 24.65) | 0.332 | 3.67 (0.19, 72.87) | **0.393** | 5.82 (1.60, 21.12) | **0.007** |
| **Pneumonia** | | | | | | | |
| No | 4.1 (2.7, 6.4) | 1.00 | | 1.00 | | 1.00 | |
| Yes | 9.7 (4.5, 20.8) | 5.54 (3.74, 8.22) | <0.001 | 3.28 (1.36, 7.86) | **0.008** | 1.99 (0.98, 4.04) | **0.056** |

*Based on complete data (N = 451). PR = prevalence ratio; CI = confidence interval.

© Adjusted for age and sex. Both unadjusted and adjusted results account for site level clustering.

Heart disease showed a consistent association with hospitalization among RSV-positive individuals. After adjustment, heart disease demonstrated a borderline association in the complete-case analysis (aPR: 3.16; 95% CI: 0.98–10.24; p = 0.054) and remained statistically significant in the imputed analysis (aPR: 2.89; 95% CI: 1.11–7.43; p = 0.030).

In contrast, neither hypertension nor diabetes was significantly associated with hospitalization in RSV-positive individuals. Hypertension showed a non-significant increase in hospitalization prevalence in both the complete-case (aPR: 1.94; 95% CI: 0.65–5.79; p = 0.234) and imputed analyses (aPR: 1.82; 95% CI: 0.66–4.95; p = 0.254). Diabetes was not associated with hospitalization in either the complete-case (aPR: 0.42; 95% CI: 0.06–3.37; p = 0.431) or imputed models (aPR: 0.49; 95% CI: 0.06–4.55; p = 0.456).

## Discussion

Over 15 years of national ILI/SARI surveillance in Uganda, RSV emerged as an important contributor to medically attended respiratory disease among adults aged ≥65 years. Its surveillance-based period prevalence was lower than SARS-CoV-2, comparable to influenza A, and higher than influenza B, underscoring the need to recognize RSV alongside other major viral pathogens in this population. Unlike the typical symptom profile of acute respiratory infections, RSV was not strongly associated with common features such as cough or sore throat. Instead, it showed strong links with asthma

**Table 5. Comorbid conditions associated with hospitalization among RSV-positive adults aged ≥65 years.**

| Comorbid condition | Hospitalization prevalence % (95% CI) | Unadjusted PR (95% CI)* | P value | Complete case Adjusted© PR (95% CI)* | P value | Imputed Adjusted© PR (95% CI) | P value |
|---|---|---|---|---|---|---|---|
| Heart disease | | | | | | | |
| No | 5.5 (3.5, 8.6) | 1.00 | | 1.00 | | 1.00 | |
| Yes | 16.7 (5.8, 48.2) | 2.67 (1.15, 9.35) | 0.022 | 2.58 (0.93, 7.14) | 0.068 | 2.88 (1.11, 7.43) | **0.029** |
| Diabetes | | | | | | | |
| No | 5.1 (3.3, 7.8) | 1.00 | | 1.00 | | 1.00 | |
| Yes | 3.2 (0.5, 22.9) | 0.52 (0.05, 5.29) | 0.580 | 0.42 (0.05, 3.89) | 0.443 | 0.49 (0.05, 4.55) | 0.531 |
| Hypertension | | | | | | | |
| No | 4.6 (2.9, 7.3) | 1.00 | | 1.00 | | 1.00 | |
| Yes | 7.7 (3.0, 19.9) | 1.42 (0.76, 2.65) | 0.273 | 1.84 (0.69, 4.91) | 0.225 | 1.81 (0.66, 4.95) | 0.251 |
| Asthma | | | | | | | |
| No | 5.6 (3.6, 8.6) | 1.00 | | 1.00 | | 1.00 | |
| Yes | 50.0 (16.1, 99.9) | 8.30 (1.61, 42.69) | 0.011 | 12.52 (1.37, 114.39) | **0.025** | 10.83 (2.25, 52.17) | **0.003** |
| Pneumonia | | | | | | | |
| No | 3.1 (1.9, 5.1) | 1.00 | | 1.00 | | 1.00 | |
| Yes | 9.8 (4.6, 21.2) | 5.01 (2.94, 8.53) | <0.001 | 3.73 (1.88, 7.41) | **<0.001** | 2.46 (0.98, 6.18) | 0.055 |

*Based on complete data (N = 466). PR = prevalence ratio; CI = confidence interval.

© *Adjusted for age and sex. Both unadjusted and adjusted results account for site level clustering.*

and pneumonia, suggesting that RSV may act as a key underlying driver of clinical complications that often necessitate hospitalization among older adults presenting for care. The circulation of RSV was also shaped by the COVID-19 pandemic: a marked suppression was observed during 2020–2021, coinciding with mitigation measures, followed by a clear re-emergence in 2022 and subsequent years.

## RSV prevalence in context

Pooled hospital-based data comprised of demographic and clinical characteristics over 15 years of sentinel surveillance indicate an RSV period prevalence among medically attended ILI/SARI cases of 4.8% among Ugandan adults aged ≥65 years, representing one of the few age-stratified estimates available from sub-Saharan Africa. This estimate lies within the 3–7% annual attack rates reported for community-dwelling older adults in high-income countries [2,12] and also comparable to hospital-based estimates in high-risk elderly subgroups with cardiopulmonary comorbidities [2,20]. However from the East African Community, a systematic review and meta-analysis of acute respiratory infection studies reported a pooled RSV prevalence of approximately 11% (95% CI 7–15), with marked differences in country-specific rates (~3% in Uganda, ~6% in Kenya, and ~29% in Tanzania), noting that estimates for Uganda and Tanzania were each derived from a single primary study, likely reflecting differences in climate, diagnostic methods, and study populations [21]. Unlike much of the earlier regional data, which often relied on antigen-based assays and aggregated all ages, our study used systematic RT-PCR testing and applied strict ILI/SARI criteria to a defined elderly cohort across multiple ecological zones. The modest period prevalence observed here likely reflects both the restricted age range and the focus on healthcare-seeking episodes, which captures more severe but fewer mild infections. Variability between our findings and pooled regional estimates highlights the importance of age-specific RSV surveillance to accurately characterise healthcare-associated disease burden, distinct from community-level incidence. Given Uganda's growing elderly population and the availability

of licensed RSV vaccines for older adults [4,22], such data are critical to inform targeted prevention strategies, optimise clinical recognition, and integrate RSV testing into existing respiratory virus surveillance platforms.

## Clinical presentation

In our elderly cohort, cough and sore throat were the only symptoms significantly associated with RSV infection, although other respiratory and systemic features such as shortness of breath, fatigue, and myalgia were also observed. While cough was reported in over 95% of RSV-positive cases, it did not remain statistically significant at multivariable analysis, suggesting that although common, it may not be a reliable discriminator of RSV infection in this population. This contrasts with reports from high-income settings where cough is considered a near-universal feature of adult RSV and is frequently accompanied by upper-respiratory symptoms such as sore throat and running nose rather than high fever [2,23].

By contrast, fever could not be meaningfully evaluated as a distinguishing clinical feature in our cohort because it was a core component of both the ILI and SARI enrollment criteria. Its near-universal presence across all participants precluded assessment of its discriminative value for RSV. Importantly, the requirement for fever as part of ILI/SARI case definitions likely resulted in under-ascertainment of afebrile RSV infections, which are increasingly recognized among older adults. Consequently, our estimates may underestimate the true healthcare-associated RSV burden in this age group, particularly for milder or atypical presentations that do not trigger care-seeking under fever-based surveillance criteria. Such afebrile presentations are biologically plausible, given the effects of immunosenescence. Age-related declines in innate and adaptive immune function blunt the production of pro-inflammatory cytokines and febrile responses, leading to attenuated systemic signs of infection [24]. Similar findings have been reported in LMIC-based surveillance studies, where reliance on fever-based case definitions has been shown to miss a substantial proportion of RSV cases among the elderly [21]. These observations underscore the need to revisit RSV case definitions for older adults and to incorporate clinical features beyond fever in order to improve case detection and surveillance accuracy.

## Role of comorbidities

Comorbidities are well-established modifiers of RSV susceptibility, clinical severity, and healthcare utilisation in older adults. In our cohort, asthma showed a strong association with RSV infection and hospitalization. However, this finding was based on a very small number of asthma cases (n = 4 overall), resulting in wide confidence intervals and statistically unstable effect estimates. Accordingly, the observed association should be interpreted cautiously and regarded as hypothesis-generating rather than definitive.

Despite this limitation, the direction of association is biologically plausible. Chronic airway inflammation, structural remodelling, and impaired mucociliary clearance in asthma may create a permissive environment for viral adherence, persistence, and heightened symptom severity [25,26]. Similar patterns have been observed in studies from North America and Asia, where asthma and chronic obstructive pulmonary disease (COPD) consistently emerge as major risk factors for RSV-related hospitalisation, prolonged recovery, and intensive care admission [27,28].

In this cohort, few RSV-positive cases were detected among patients with heart disease, likely reflecting under-reporting of comorbidities in routine sentinel surveillance data. Although the prevalence among cases with heart disease was modest, this remains notable given global evidence linking pre-existing cardiovascular disease to poorer RSV outcomes. Pathophysiologic mechanisms include reduced cardiopulmonary function, RSV-triggered myocardial stress, and systemic inflammation that can precipitate acute decompensation [29]. These findings support the prioritizing of older adults with chronic respiratory or cardiovascular disease in RSV prevention strategies, including vaccination and early antiviral interventions where available. Strengthening comorbidity data capture in surveillance systems will improve risk stratification, while integrated care pathways may help mitigate the disproportionate RSV healthcare burden in these high-risk groups.

## Coinfection transmission patterns

In this elderly cohort, all viruses co-circulated year-round, but RSV–influenza coinfections were rare (0.4%), and no RSV–SARS-CoV-2 coinfections were detected. These findings are consistent with both high- and low-income country data, where dual RSV–SARS-CoV-2 infections remain infrequent and influenza–RSV co-detection in adults typically falls below 1% [30,31]. However, the absence of detected RSV–SARS-CoV-2 coinfections in this study is likely attributable to methodological and epidemiological constraints rather than true biological absence. Specifically, the relatively small number of RSV-positive cases limited statistical power to detect rare dual infections, even during periods of documented co-circulation. In addition, asynchronous epidemic peaks of RSV and SARS-CoV-2—particularly in the post-pandemic period—may have reduced temporal overlap at the individual level. Finally, surveillance relied on single respiratory specimens collected per illness episode, which may have missed sequential or short-lived infections occurring outside the sampling window. Accordingly, the lack of observed RSV–SARS-CoV-2 coinfections should not be interpreted as evidence that such coinfections do not occur in older adults.

Biologically, such transmission dynamics may be explained through innate immune response whereby by viral interference, whereby infection with one respiratory virus induces an antiviral state through type I and III interferon responses that suppress replication of other viruses [32]. Although innate immune responses may limit opportunities for viral co-infection [33], asynchronous epidemic peaks (particularly in East Africa, where RSV often peaks before influenza) can result in protracted outbreaks, with one virus dominating a season and another emerging sequentially.

Pandemic-era public health measures, including masking, physical distancing, and travel restrictions, likely further reduced simultaneous viral transmission, as observed in global surveillance reports [34]. Although the limited number of coinfections in our dataset precluded formal outcome comparisons, previous studies suggest that RSV monoinfection alone can cause substantial morbidity in older adults, with coinfection adding little or only modestly to clinical severity in this age group [4].

Continued multi-pathogen testing in sentinel surveillance systems remains important to detect shifts in these patterns, particularly as COVID-19 control measures are relaxed and respiratory virus seasonality potentially re-aligns, increasing the theoretical window for coinfections.

## Seasonal and pandemic-related dynamics

RSV epidemics demonstrate marked geographic variation, shaped by the interplay of climate, viral ecology, and host susceptibility. In our dataset, peaks occurred consistently during May–June, coinciding with the end of Uganda's long rainy season and the beginning of the short dry season; a pattern also observed in neighboring Kenya [35]. This contrasts with temperate regions, where RSV activity typically peaks in winter months, underscoring the influence of local climatic drivers on epidemic timing [36,37]. In tropical East Africa, high humidity may stabilise virions and facilitate transmission, while seasonal rainfall may increase indoor crowding, amplifying spread [37]. However, when aggregated by rainfall period, RSV prevalence did not differ significantly between wet and dry seasons, suggesting largely year-round transmission with only modest seasonal modulation in older adults. This likely reflects heterogeneous healthcare-seeking behavior and the focus on medically attended ILI/SARI episodes rather than community incidence.

The near-complete absence of RSV circulation in 2020–2021 parallels global reports of disrupted RSV seasonality during the COVID-19 pandemic, when non-pharmaceutical interventions (including masking, school closures, and mobility restrictions) suppressed respiratory virus transmission [34]. The subsequent resurgence observed from 2022–2023 is consistent with the "immunity debt" hypothesis, whereby prolonged suppression of viral circulation during periods of intensive non-pharmaceutical interventions leads to an accumulation of susceptible individuals and larger or atypically timed outbreaks once interventions are relaxed. This concept has been supported by epidemic modelling studies, including Baker et al., who demonstrated that COVID-19 mitigation measures substantially reduced RSV transmission and predicted increased outbreak magnitude and altered timing in subsequent seasons due to rising

population-level susceptibility [38]. Similar post-intervention rebounds in RSV activity have since been documented across multiple global settings [39,40].

Taken together, the non-significant seasonality observed in this surveillance cohort likely reflects standardized, year-round enrollment across sites and seasons, coupled with persistent RSV circulation in tropical settings. These findings reinforce the importance of continuous, year-round RSV surveillance in LMICs to detect shifts in timing and intensity as climate variability, urbanization, and post-pandemic behavioral changes continue to reshape transmission patterns.

## Public health and clinical implications

Our findings confirm RSV as a persistent contributor to acute respiratory illness and hospitalization among medically attended ILI/SARI episodes in older Ugandan adults, despite its lower prevalence relative to SARS-CoV-2. Importantly, the estimates presented here reflect healthcare-associated RSV burden captured through sentinel surveillance, rather than population-level incidence, and should be interpreted within the context of fever-based case definitions that may under-ascertain milder or afebrile infections. By demonstrating strong associations with asthma and other chronic respiratory conditions, this study highlights the importance of prioritising high-risk groups (particularly older adults with chronic lung or cardiovascular disease) for preventive interventions once vaccines or monoclonal prophylaxis become available in LMICs. Importantly, the concentration of RSV burden among adults aged ≥65 years in this study directly supports this age group as an appropriate and evidence-based target for RSV prevention strategies, including vaccination, in Uganda and similar settings.

For clinicians, heightened suspicion of RSV during seasonal peaks is essential, given its nonspecific presentation and overlap with influenza, COVID-19, and bacterial pneumonia. Incorporating RSV testing into diagnostic algorithms for older adults could improve case recognition, reduce inappropriate antibiotic use, and contribute to antimicrobial resistance mitigation. From a public health perspective, the seasonal concentration of cases suggests that future vaccination campaigns in Uganda could be optimally timed to precede the long rainy season. Age-targeted vaccination approaches focusing on adults ≥65 years, particularly those with underlying cardiopulmonary disease, may maximise public health impact while remaining programmatically feasible in resource-constrained settings. However, variability in climate and epidemic timing reinforces the need for sustained, year-round surveillance to guide implementation.

Beyond epidemic timing, continuous RSV surveillance will be critical for detecting post-pandemic shifts in circulation, monitoring the emergence of vaccine-escape variants, and informing cost-effectiveness models for preventive strategies. The age-specific estimates generated by this study provide essential baseline data for evaluating the potential impact, prioritisation, and cost-effectiveness of RSV vaccination strategies targeting older adults in sub-Saharan Africa. Integration of RSV testing into existing influenza and COVID-19 surveillance systems offers a pragmatic and resource-efficient path forward. Such measures would provide actionable evidence for policy, enhance preparedness for an aging population, and reduce the growing burden of respiratory morbidity in sub-Saharan Africa.

## Strengths and limitations

This study has several notable strengths. It represents the most comprehensive analysis to date of RSV epidemiology among older adults in Uganda, and one of the few from sub-Saharan Africa. The 15-year period of continuous surveillance (2010–2025) enabled assessment of RSV patterns before, during, and after the COVID-19 pandemic, thereby capturing both long-term seasonal dynamics and pandemic-related disruptions. Inclusion of sentinel sites across diverse ecological zones—urban, peri-urban, and rural—enhanced representativeness for Uganda's older adult population. Laboratory confirmation of RSV, influenza A/B, and SARS-CoV-2 using real-time RT-PCR ensured high diagnostic accuracy, while the collection of detailed demographic, clinical, and comorbidity data facilitated multivariable analyses of predictors of infection and hospitalization.

However, several limitations should be considered when interpreting these findings. A key limitation is the small absolute number of RSV-positive cases identified over the 15-year surveillance period (n = 26), corresponding to an average of fewer than one RSV case per week across all participating sites. This reflects both the relatively low prevalence of RSV among medically attended ILI/SARI episodes in older adults and the surveillance design, which captures only healthcare-seeking individuals meeting strict case definitions. The limited number of RSV cases reduced statistical power, constrained subgroup analyses, and contributed to wide confidence intervals for some effect estimates, particularly for comorbidities such as asthma. Given current surveillance targets of approximately 50 enrollments per week across all ages, these findings suggest that only a small fraction of enrolled cases are likely to be adults aged ≥65 years; future national RSV prevention strategies may therefore require targeted oversampling or age-prioritized enrollment to achieve sufficient sample sizes for robust evaluation in this high-risk group.

Second, the cross-sectional design, although based on long-term surveillance, limits causal inference regarding the temporal relationship between exposures (e.g., comorbidities) and RSV outcomes. Third, hospitalization was used as a proxy measure for severe disease; while pragmatic in a surveillance context, it may be influenced by health-seeking behavior, bed availability, and clinician admission practices, potentially leading to misclassification of severity. Fourth, pneumonia status was recorded as a clinician-diagnosed condition based on routine clinical assessment documented on sentinel case investigation forms. Because systematic radiographic confirmation and standardized diagnostic coding were not uniformly available across sites and years, diagnostic variability and misclassification of pneumonia severity may have occurred.

Fifth, comorbidity data relied on clinical records, which may underreport chronic conditions, particularly in settings with limited diagnostic capacity. Sixth, certain variables, including hospitalization status, had notable missingness, and although missingness was generally unrelated to RSV status, it could have reduced statistical power in some analyses. Finally, the absence of virological subtyping and viral load data precluded exploration of potential differences in clinical presentation or severity by RSV subtype (A vs. B).

Taken together, these limitations indicate that associations observed in this study, especially those based on small numbers, should be interpreted cautiously and viewed as hypothesis-generating. Despite these limitations, the study provides robust, nationally representative, and long-term evidence on RSV in older Ugandan adults. Future research should incorporate longitudinal cohort designs, improved comorbidity ascertainment, and molecular characterization of RSV strains to deepen understanding of clinical and epidemiological heterogeneity.

## Conclusions

This 15-year sentinel surveillance analysis is the most comprehensive evaluation of RSV in older adults in Uganda. RSV accounted for a consistent fraction of respiratory illness, comparable to influenza A, with a distinct seasonal peak during the rainy season. Asthma, pneumonia, and heart disease were key predictors of severe illness, while coinfections were rare. Pandemic-era suppression and later resurgence further illustrate the dynamic nature of RSV transmission. By filling an evidence gap in an underrepresented, high-risk population, this study supports integrating RSV into national surveillance frameworks, prioritising high-risk groups for prevention, and sustaining year-round monitoring to guide timely vaccine introduction.

## Supporting information

**S1 Dataset. Minimal dataset underlying the analyses reported in the manuscript.** This file contains the de-identified raw data used to generate all summary statistics, tables, and figures in the study, including the values underlying reported means, standard deviations, proportions, and graphical outputs. Variable definitions and coding are provided within the file to enable full replication of the analyses.
(XLSX)

## Acknowledgments

This work leveraged the Interdisciplinary Consortium for Epidemics Research (ICER) platform. We gratefully acknowledge the contributions of ICER in fostering collaboration across disciplines and providing critical support for advancing research on epidemic preparedness, response, and recovery.

## Author contributions

**Conceptualization:** Haruna Muwonge, Julius Lutwaama, Barnabas Bakamutumaho.

**Data curation:** Haruna Muwonge, Joyce Namulondo, Levicatus Mugenyi, Joweria Nakaseegu, Bridget Nakamoga, Esther Amwine, David Odongo, John Kayiwa, Barnabas Bakamutumaho.

**Formal analysis:** Levicatus Mugenyi, David Odongo, Barnabas Bakamutumaho.

**Funding acquisition:** Haruna Muwonge, Barnabas Bakamutumaho.

**Investigation:** Haruna Muwonge, Joyce Namulondo, Joweria Nakaseegu, Esther Amwine, John Kayiwa, Barnabas Bakamutumaho.

**Methodology:** Haruna Muwonge, Bridget Nakamoga, Barnabas Bakamutumaho.

**Project administration:** Haruna Muwonge, Julius Lutwaama, Bruce Kirenga.

**Resources:** Haruna Muwonge, Julius Lutwaama, Bruce Kirenga.

**Supervision:** Haruna Muwonge, Bruce Kirenga, Barnabas Bakamutumaho.

**Validation:** Haruna Muwonge.

**Visualization:** Haruna Muwonge, Levicatus Mugenyi.

**Writing – original draft:** Haruna Muwonge, Barnabas Bakamutumaho.

**Writing – review & editing:** Haruna Muwonge, Joyce Namulondo, Levicatus Mugenyi, Joweria Nakaseegu, Bridget Nakamoga, Esther Amwine, Roselyne Akugizibwe, Abdul Ssekandi, Mustafa Ssaka, Hassan Kasujja, Godfrey S Bbosa, David Odongo, Julius Lutwaama, John Kayiwa, Bruce Kirenga, Barnabas Bakamutumaho.

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
