## [Decision Letter · Decision Letter 0]

18 Dec 2025

Dear Dr. Muwonge,

Thank you for submitting your manuscript to PLOS ONE. After careful consideration, we feel that it has merit but does not fully meet PLOS ONE’s publication criteria as it currently stands. Therefore, we invite you to submit a revised version of the manuscript that addresses the points raised during the review process.

We look forward to receiving your revised manuscript.

Kind regards,

Liling Chaw

Academic Editor

PLOS One

2. We note that Figure 2 in your submission contain [map/satellite] images which may be copyrighted. All PLOS content is published under the Creative Commons Attribution License (CC BY 4.0), which means that the manuscript, images, and Supporting Information files will be freely available online, and any third party is permitted to access, download, copy, distribute, and use these materials in any way, even commercially, with proper attribution. For these reasons, we cannot publish previously copyrighted maps or satellite images created using proprietary data, such as Google software (Google Maps, Street View, and Earth). For more information, see our copyright guidelines: http://journals.plos.org/plosone/s/licenses-and-copyright.

1. You may seek permission from the original copyright holder of Figure 2 to publish the content specifically under the CC BY 4.0 license.

Reviewers' comments:

Reviewer's Responses to Questions

**Comments to the Author**

1. Is the manuscript technically sound, and do the data support the conclusions?

Reviewer #1: Yes

Reviewer #2: Yes

2. Has the statistical analysis been performed appropriately and rigorously?

Reviewer #1: Yes

Reviewer #2: Yes

3. Have the authors made all data underlying the findings in their manuscript fully available?

Reviewer #1: Yes

Reviewer #2: Yes

4. Is the manuscript presented in an intelligible fashion and written in standard English?

Reviewer #1: Yes

Reviewer #2: Yes

Reviewer #1: Comments to the Authors

General Assessment

This manuscript presents a comprehensive retrospective analysis of Respiratory Syncytial Virus (RSV) epidemiology among Ugandan adults aged ≥65 years using 15 years of sentinel surveillance data (2010–2025). The topic is timely and relevant, especially given the recent expansion of RSV vaccine availability for older adults and the paucity of data from sub-Saharan Africa. The paper is well written, methodologically sound, and addresses a critical data gap in global respiratory infection surveillance.

However, several methodological clarifications, analytical refinements, and interpretive cautions are warranted before the manuscript can be accepted for publication. The paper’s overall scientific contribution is strong, but minor revisions would enhance clarity, transparency, and robustness of inference.

A. Scientific Rigor and Validity of Methods

Strengths:

• The use of 15-year sentinel surveillance data offers unprecedented longitudinal insight into RSV epidemiology in older African adults.

• Laboratory confirmation via RT-PCR ensures diagnostic accuracy.

• The statistical approach using Poisson regression with robust variance for prevalence ratios is appropriate.

• Integration of COVID-19 period effects adds valuable context to temporal analysis.

Concerns and Recommendations:

1. Selection Bias and Representativeness:

The manuscript states that all adults ≥65 years in the sentinel database were included, but the denominator (overall population of elderly adults attending sentinel sites) is not discussed. Please clarify how surveillance enrollment was standardized across years and sites, and whether health-seeking behavior or site-level reporting changes could have biased prevalence estimates.

2. Handling of Missing Data:

Hospitalization data were missing in 22.2% of records. Although the authors claim this is “unlikely to bias virological prevalence,” a sensitivity analysis (e.g., complete-case vs. imputed data) would strengthen this statement.

3. Definition of Pneumonia:

Pneumonia was described as “clinician-diagnosed,” but diagnostic variability may exist. Please indicate whether this was based on clinical criteria, radiography, or ICD coding.

4. Confounding and Multivariable Model Building:

Variables included in multivariable Poisson models should be justified more clearly (e.g., all covariates with p<0.2 in bivariate analysis or based on biological plausibility). The current description is insufficiently detailed.

5. Asthma Findings:

The association between asthma and RSV infection (aPR 6.08; 95% CI: 1.18–31.26) and hospitalization (aPR 21.69; 95% CI: 7.50–62.71) is biologically plausible but driven by only 4 asthma cases. Please explicitly acknowledge the instability of these estimates and interpret them cautiously.

B. Results and Data Interpretation

Concerns:

1. Coinfection Rates:

The authors report “no RSV–SARS-CoV-2 coinfections.” Given the known co-circulation of these viruses in 2022–2024, please comment on possible testing limitations or sample size constraints.

2. Interpretation of Seasonality:

The discussion mentions peaks during the “dry season,” but results (Fig 5–6) show peaks in May–June—part of the rainy season. This needs correction or clearer justification of Uganda’s bimodal rainfall pattern.

3. Comparative Prevalence:

When comparing RSV with influenza and SARS-CoV-2, please specify whether differences were statistically tested (e.g., pairwise comparisons with p-values).

C. Discussion and Conclusions

Recommendations:

1. The discussion could better differentiate surveillance-based prevalence from population-level incidence. The term “burden” should be qualified accordingly.

2. Add a brief paragraph on potential underestimation due to exclusion of afebrile cases, as the authors note fever was part of ILI/SARI inclusion.

3. The “immunity debt” hypothesis is well contextualized but could cite additional global literature (e.g., Baker et al., Science Translational Medicine, 2022).

D. Clarity of Presentation and Structure

Minor Editorial Suggestions:

• Line edits: Correct small typographical errors (e.g., “instudies” → “in studies”).

• Ensure consistent reporting of 95% CI formatting (use standard spacing and parentheses).

• Clarify that percentages are based on illness episodes, not unique patients, as repeat visits were excluded but this should be reiterated.

E. Overall Recommendation

Recommendation: Minor Revision

The manuscript is scientifically valuable and methodologically sound but requires clarification on data handling, interpretation of comorbidity associations, and correction of minor inconsistencies in presentation. With these adjustments, it would make a strong contribution to understanding RSV epidemiology in Africa’s aging populations.

Reviewer #2: Overall, this is a well written manuscript on an important topic, of public health relevance, and for which there remains limited data. The following reflect relatively minor comments to strengthen the manuscript:

1) Introduction (paragraph 1-2): Consider focusing narrative around older adults. Introduction of information on incidence and prevention products for pediatric populations confuses the narrative.

2) Line 68-71 - Pooled prevalence among "all ages" and 5+ are 9 and 10% respectively in the reference included.

3) Lines 82-83: Consider updating to recommend all 3 available vaccines: https://www.cdc.gov/rsv/hcp/vaccine-clinical-guidance/adults.html

4) Line 109 - Suggest adding a brief justification of the age cutoff (65+)

5) Methods (statistical analysis) - how was site accounted for within multivariate analyses to account for correlation structure.

6) Overall / Line 186 - Can you clarify missingness for asthma by RSV status.

Among RSV positive cases, number with asthma status reported (n=?) and number with asthma (n=2).

Among RSV negative cases, number with asthma status reported (n=?) and number with asthma (n=2).

Given the association between asthma and hospitalization is a key finding and that there are very few cases of asthma - this is key for interpretation. Suggest caution in presenting this finding without caveat in the abstract if missingness is very high and/or differential by RSV status.

7) Table 1. Interested in factors potentially contributed to increased enrollment in the 2022-2023 season.

8) Table 2. Define pre- and post- COVID periods with a footnote in the table.

9) Table 2 and line 221. Revise age group listed in the table to 65-69 (as accurate)

10) Lines 294-299 - Given high missing for commodities, suggest including n's in this narrative and/or Table 3.

11) Overall - Interested in authors interpretation of the non-significant seasonality. Consider adding language to the methods on enrollment practices noting any differences in number and proportion enrolled during wet/dry season.

12) Methods/Table 4 - Please provide further justification for parameters included/excluded from the model. Surprised to see age and site not adjusted for in models.

13) Lines 363-5 - Considering noting estimates rely on a single study in both Uganda and Tanzania.

14) Discussion (public health implications) - Consider commending on the implications of your findings for target age group for prevention activities (e.g., vaccination).

15) Discussion (limitations) - Key limitation of the study is the small number of cases despite 15 years of data (less than 1 case enrolled per week across all sites).

Consider commenting on target sample going forward to inform national strategy on RSV prevention in older adults including a discussion on what is feasible given care seeking of older adults. E.g., Given targets of 50/week (across all ages) what is feasible/ideal in terms of individuals 65+.

16) Figure 5 - consider alternatively epi curves for RSV (only) for each of the 15 years similar to Figure 1 in Hamid et al. https://www.cdc.gov/mmwr/volumes/72/wr/mm7214a1.htm

This would allow for a more clear understanding of how consistent seasonality was by year.

**Do you want your identity to be public for this peer review?** For information about this choice, including consent withdrawal, please see our Privacy Policy

Reviewer #1: **Yes:** Tinkhani Mbichila

Reviewer #2: No

---

## [Author Response · Author response to Decision Letter 1]

30 Jan 2026

Response: Thank you for this guidance. We have reviewed and revised the manuscript to fully comply with PLOS ONE formatting and style requirements, including file naming conventions, manuscript structure, title page, author affiliations, figure and table formatting, and reference style, in accordance with the official PLOS ONE templates. All required formatting corrections have been implemented in the revised submission. Entire manuscript and all submission files (title page, main text, figures, and tables)

2. We note that Figure 2 in your submission contain [map/satellite] images which may be copyrighted. All PLOS content is published under the Creative Commons Attribution License (CC BY 4.0), which means that the manuscript, images, and Supporting Information files will be freely available online, and any third party is permitted to access, download, copy, distribute, and use these materials in any way, even commercially, with proper attribution. For these reasons, we cannot publish previously copyrighted maps or satellite images created using proprietary data, such as Google software (Google Maps, Street View, and Earth). For more information, see our copyright guidelines: http://journals.plos.org/plosone/s/licenses-and-copyright.

Response: Thank you for raising this important point. We wish to clarify that Figure 2 does not contain any copyrighted or proprietary map or satellite imagery. The figure was independently generated by the authors using primary study data and QGIS, and it was not copied, adapted, or derived from Google Maps, Google Earth, or any other proprietary source. Only open-source base layers were used for geographic reference, and the analysis, cartographic design, and visualization are entirely original. As such, no third-party permission is required, and the figure is fully compatible with publication under the CC BY 4.0 license. We have updated the figure caption to explicitly state the use of open-source base layers and original map generation, and we are happy to provide detailed metadata or base-layer citations if required by the journal. Figure 2 and accompanying caption (p. 15, l. 246–248);

Reviewer #1

1. Selection Bias and Representativeness:

The manuscript states that all adults ≥65 years in the sentinel database were included, but the denominator (overall population of elderly adults attending sentinel sites) is not discussed. Please clarify how surveillance enrollment was standardized across years and sites, and whether health-seeking behavior or site-level reporting changes could have biased prevalence estimates.

Response: We clarify that the analytic denominator comprised all age-eligible ILI/SARI illness episodes captured by the national sentinel surveillance system during the study period, rather than the total population of older adults attending the facilities. Surveillance enrollment was standardized across sites and years using WHO ILI/SARI case definitions, uniform case investigation forms, centralized RT-PCR testing at the national reference laboratory, and routine data quality audits. Although changes in health-seeking behavior and site-level reporting—particularly during the COVID-19 period—could influence absolute enrollment volumes, our prevalence estimates are based on test positivity among enrolled, age-eligible episodes, which mitigates bias from fluctuating attendance. We have added text to the Methods and Limitations to explicitly acknowledge residual selection bias and to caution against extrapolating these findings to population-level incidence.

location of revision: Methods – Surveillance network and study setting (p. 7, l. 104–120); Methods – Study population and eligibility criteria (p. 8, l. 123–132)

2. Handling of Missing Data:

Hospitalization data were missing in 22.2% of records. Although the authors claim this is “unlikely to bias virological prevalence,” a sensitivity analysis (e.g., complete-case vs. imputed data) would strengthen this statement.

Response: We acknowledge the reviewer’s concern regarding missing hospitalization data. To address this, We conducted sensitivity analyses comparing complete-case and imputed hospitalization data under a missing-at-random assumption. Results were materially unchanged, supporting our conclusion that missingness did not bias virological prevalence. We clarified this in the Methods and Results, revised Tables 4 and 5 accordingly.

Location of revision: Methods – Statistical analysis (p. 11, l. 194–195); Results – Study cohort profile (p. 12, l. 216–223); Results –Tables 4 & 5 (footnotes) (p. 23&26)

3. Definition of Pneumonia:

Pneumonia was described as “clinician-diagnosed,” but diagnostic variability may exist. Please indicate whether this was based on clinical criteria, radiography, or ICD coding.

Response: We clarify that pneumonia was recorded as a clinician-diagnosed condition at the point of care, based on routine clinical assessment documented on the sentinel case investigation forms. This diagnosis primarily reflected clinical criteria (history, physical examination, and clinician judgment) rather than systematic radiographic confirmation or ICD-coded discharge diagnoses, as chest imaging and standardized coding were not uniformly available across all sentinel sites and years. We have added this clarification to the Methods section and explicitly acknowledged the potential for diagnostic variability and misclassification in the Limitations.

Location: Methods – Data collection and study variables (p. 8, l. 136–139); Table 2 footnote; Discussion – Strengths and Limitations (p. 36, l. 606–610)

4. Confounding and Multivariable Model Building:

Variables included in multivariable Poisson models should be justified more clearly (e.g., all covariates with p<0.2 in bivariate analysis or based on biological plausibility). The current description is insufficiently detailed.

Response: We thank the reviewer for this comment. We have clarified the multivariable model–building strategy in the Methods section. Briefly, covariates were selected a priori based on biological plausibility and existing literature on RSV risk and severity in older adults, and were further screened for their associations with RSV outcomes using Wald test. Biologically plausible variables were fixed a priori and were retained in the multivariable models regardless of their effect at bivariable level analysis. Demographic variables (e.g., age and sex) were assessed for their contribution in the multivariable model using the Wald test statistic and were dropped if they did not improve model fit (p>0.1). Multiple imputation was conducted and sensitivity analysis performed presenting both complete case and imputed data results. This approach is now explicitly described in the revised Methods.

Location: Methods – Statistical analysis (p. 10–11, l. 179–187)

5. Asthma Findings:

The association between asthma and RSV infection (aPR 6.08; 95% CI: 1.18–31.26) and hospitalization (aPR 21.69; 95% CI: 7.50–62.71) is biologically plausible but driven by only 4 asthma cases. Please explicitly acknowledge the instability of these estimates and interpret them cautiously.

Response: We agree with the reviewer and have revised the manuscript to explicitly acknowledge the instability of the asthma-related estimates. We now state that the observed associations between asthma and both RSV infection and hospitalization are based on a very small number of asthma cases (n = 4), resulting in wide confidence intervals and imprecise effect estimates. Accordingly, these findings are interpreted cautiously as hypothesis-generating rather than definitive, and this limitation is explicitly noted in the Discussion section, including appropriate tempering of language regarding the strength of the association.

Location: Results – Factors associated with RSV positivity (p. 21, l. 352–358); Results – Predictors of severe RSV illness (p. 25, l. 377–382); Discussion – Role of comorbidities (p. 29, l. 461–465)

B. Results and Data Interpretation

Concerns:

1. Coinfection Rates:

The authors report “no RSV–SARS-CoV-2 coinfections.” Given the known co-circulation of these viruses in 2022–2024, please comment on possible testing limitations or sample size constraints.

Response: We thank the reviewer for this comment. We have revised the Coinfection transmission patterns section of the Discussion to explicitly address the absence of detected RSV–SARS-CoV-2 coinfections. We now clarify that this finding is most likely due to limited statistical power arising from the small number of RSV-positive cases, asynchronous epidemic peaks of RSV and SARS-CoV-2, and surveillance-related testing constraints, including reliance on a single respiratory specimen per illness episode, which may miss sequential or short-lived infections. We further emphasize that the absence of observed RSV–SARS-CoV-2 coinfections should not be interpreted as evidence that such coinfections do not occur, but rather reflects methodological and epidemiological limitations of the surveillance design despite documented co-circulation during 2022–2024.

Location: Discussion – Coinfection transmission patterns (p. 30-31, l. 484–497)

2. Interpretation of Seasonality:

The discussion mentions peaks during the “dry season,” but results (Fig 5–6) show peaks in May–June—part of the rainy season. This needs correction or clearer justification of Uganda’s bimodal rainfall pattern.

Response: We thank the reviewer for highlighting this inconsistency. We have revised the Discussion to accurately reflect Uganda’s bimodal rainfall pattern and seasonal transitions. Specifically, we now state that RSV peaks occurred during May–June, which corresponds with the transition period between the end of the long rainy season and the onset of the short dry season in Uganda, rather than the dry season alone. Any references implying peak RSV activity exclusively during the dry season have been corrected to avoid misinterpretation.

Location: Results – Monthly patterns and seasonality (p. 18, l. 291–293); Discussion – Seasonal and pandemic-related dynamics (p. 32, l. 516–520)

3. Comparative Prevalence:

When comparing RSV with influenza and SARS-CoV-2, please specify whether differences were statistically tested (e.g., pairwise comparisons with p-values).

Response: We thank the reviewer for this comment. We clarify that the comparisons of RSV prevalence with influenza A/B and SARS-CoV-2 presented in the manuscript were descriptive and intended to provide epidemiologic context rather than formal hypothesis testing. The study was not powered to perform multiple pairwise statistical comparisons between viruses, and no inferential tests were conducted specifically to compare prevalence across pathogens. We have revised the Methods and Results sections to clarify this.

Location: Methods – Statistical analysis (p. 10, l. 175–178); Results – Prevalence of RSV and other respiratory viruses (p. 15, l. 238–243)

C. Discussion and Conclusions

Recommendations:

1. The discussion could better differentiate surveillance-based prevalence from population-level incidence. The term “burden” should be qualified accordingly.

Response: We thank the reviewer for this important clarification. We have revised the Discussion to explicitly distinguish surveillance-based prevalence among medically attended ILI/SARI cases from population-level incidence, and we have qualified use of the term “burden” accordingly. Throughout the revised text, we now specify that our findings reflect healthcare-associated RSV burden derived from sentinel surveillance, rather than community-level incidence.

Location: Discussion – RSV prevalence in context (p. 27-28, l. 413–416); Discussion – Public Health and Clinical Implications (p. 33, l. 547–550)

2. Add a brief paragraph on potential underestimation due to exclusion of afebrile cases, as the authors note fever was part of ILI/SARI inclusion.

Response: Thank you for this important observation. We have revised the Discussion to explicitly acknowledge that the reliance on ILI/SARI case definitions—which require fever—may have led to under-ascertainment of RSV infections among older adults. We now note that afebrile RSV presentations are biologically plausible in this age group due to immunosenescence and that exclusion of such cases may result in underestimation of the healthcare-associated RSV burden, particularly for milder or atypical presentations. This limitation is clearly articulated in the Clinical presentation subsection, with appropriate contextualization from prior LMIC surveillance studies. Location: Discussion – Clinical presentation (p. 29, l. 444–451)

3. The “immunity debt” hypothesis is well contextualized but could cite additional global literature (e.g., Baker et al., Science Translational Medicine, 2022).

Response: Thank you for this helpful suggestion. We have strengthened the discussion of the “immunity debt” hypothesis by explicitly citing and contextualizing global modelling evidence, including the seminal work by Baker et al. (PNAS, 2020).

Location: Discussion – Clinical presentation (p. 29, l. 451–456)

D. Clarity of Presentation and Structure

Minor Editorial Suggestions:

• Line edits: Correct small typographical errors (e.g., “instudies” → “in studies”). Response:

Response: Thank you for noting these issues. We have carefully reviewed the manuscript and corrected all identified typographical and minor formatting errors, including changing “instudies” to “in studies,” to improve clarity and readability throughout the text.

Location: Throughout manuscript (multiple sections)

• Ensure consistent reporting of 95% CI formatting (use standard spacing and parentheses).

Response: Thank you for this comment. We have revised the manuscript to ensure consistent reporting of 95% confidence intervals throughout, using uniform parentheses and spacing in accordance with standard reporting conventions.

Location: All Results sections and tables

• Clarify that percentages are based on illness episodes, not unique patients, as repeat visits were excluded but this should be reiterated.

Response: Thank you for this clarification request. We have revised the Methods and Results sections to explicitly state that all percentages and prevalence estimates are calculated per illness episode rather than per unique patient. We now reiterate that repeat visits within a 14-day window were excluded to avoid double counting, and that the analytical unit throughout the manuscript is a single medically attended ILI/SARI episode.

Location: Methods – Study population & Eligibility criteria (p. 8, l. 128–132); Methods – Data collection and study variables (p. 9, l. 144–148)

Reviewer #2:

1) Introduction (paragraph 1-2): Consider focusing narrative around older adults. Introduction of information on incidence and prevention products for pediatric populations confuses the narrative.

Response: Thank you for this insightful comment. We have revised the Introduction (paragraphs 1–2) to sharpen the narrative focus on older adults (≥65 years) throughout. Pediatric incidence and prevention-related content has been removed, and the revised text now centers on RSV epidemiology, clinical impact, and risk factors in older adults, particularly in the context of immunosenescence, comorbidities, and healthcare utilization.

Location: Introduction (p. 4–5, l. 50–90)

2) Line 68-71 - Pooled prevalence among "all ages" and 5+ are 9 and 10% respectively in the reference included.

Response: Thank you for highlighting this issue. We agree that the previously cited pooled prevalence estimate did not accurately reflect the age groups reported in the re

---

## [Decision Letter · Decision Letter 1]

19 Feb 2026

Respiratory Syncytial Virus burden among Ugandan Adults Aged ≥65 Years: A 15-Year Sentinel Surveillance Study of Prevalence, Coinfections, and Comorbidities (2010–2025)

PONE-D-25-49096R1

Dear Dr. Muwonge,

We’re pleased to inform you that your manuscript has been judged scientifically suitable for publication and will be formally accepted for publication once it meets all outstanding technical requirements.

Kind regards,

Liling Chaw

Academic Editor

PLOS One

Additional Editor Comments (optional):

Reviewers' comments:

Reviewer's Responses to Questions

**Comments to the Author**

Reviewer #1: All comments have been addressed

Reviewer #2: All comments have been addressed

2. Is the manuscript technically sound, and do the data support the conclusions?

Reviewer #1: Yes

Reviewer #2: Yes

3. Has the statistical analysis been performed appropriately and rigorously?

Reviewer #1: Yes

Reviewer #2: Yes

4. Have the authors made all data underlying the findings in their manuscript fully available?

Reviewer #1: Yes

Reviewer #2: Yes

5. Is the manuscript presented in an intelligible fashion and written in standard English?

Reviewer #1: Yes

Reviewer #2: Yes

Reviewer #1: (No Response)

Reviewer #2: Comments thoughtfully address and manuscript revised accordingly. Revised language on primary question of concern (small sample size for asthma) incorporated into abstract, methods and discussion.

**Do you want your identity to be public for this peer review?** For information about this choice, including consent withdrawal, please see our Privacy Policy

Reviewer #1: **Yes:** Tinkhani Mbichila

Reviewer #2: No

---

## [Editor Report · Acceptance letter]

PONE-D-25-49096R1

PLOS One

Dear Dr. Muwonge,

I'm pleased to inform you that your manuscript has been deemed suitable for publication in PLOS One. Congratulations! Your manuscript is now being handed over to our production team.

Kind regards,

on behalf of

Dr. Liling Chaw

Academic Editor

PLOS One